# Nanoscale grinding: Unlocking the nutrient potential of oxidized phosphate rocks for sustainable fertilizer innovation

Houda A. Khedr [1]*, Mohamed O. Ebraheem[1], Hussah A. Alshwyeh[2,3], Najla F. Gumaah[4], Saedah Rwede AL-Mhyawi[5], Ahmed H. Ragab[6], Ahmed M. Zayed[7]*

**1** Geology Department, Faculty of Science, New Valley University, New Valley, Egypt, **2** Biology Department, College of Science, Imam Abdulrahman Bin Faisal University, Dammam, Saudi Arabia, **3** Basic and Applied Scientific Research Center (BASRC), Imam Abdulrahman Bin Faisal University, Dammam, Saudi Arabia, **4** Chemistry Department, Faculty of Science, Northern Border University, Saudia Arabia, **5** Department of Chemistry, College of Science, University of Jeddah, Jeddah, Saudi Arabia, **6** Chemistry Department, College of Science, King Khalid University, Abha, Saudi Arabia, **7** Applied Mineralogy and Water Research Lab (AMWRL), Geology Department, Faculty of Science, Beni-Suef University, Beni Suef, Egypt

\* zayed_2000eg@yahoos.com, ahmed.zayed@science.bsu.edu.eg (AMZ); houda.mohamed238@gmail.com (HAK)

## Abstract

The current study delves into the transformative effects of intensive grinding to nanoscale upon oxidized phosphate rocks (PRs) of various grades, high (HMP), medium (MMP) and low (LMP) micro-sizes. Hence, the consequences of these transformative changes on phosphorous dissolution rate of these fractions using acetic acid, were carefully evaluated. The produced high (HNP) and medium (MNP) grades of nano-sized fractions revealed significant changes in their chemical composition, mineralogical, morphological and geometrical properties. Whereas the low grade, LNP, was moderately changed. HNP and MNP exhibited a remarkable increase in structural disorder (slight broadening of reflections) and Loss on Ignition (LOI) contents (10.62 and 13 wt.%, orderly), surpassing their counterparts (HMP: 6.04 and MMP: 10.92 wt.%). Despite the reduction in their $P_2O_5$ contents, HNP (31.23 wt.% and MNP (24.22 wt.%), astoundingly outperformed their micro-sized equivalents (HMP: 35.70 wt.%, MMP: 27.92 wt.%) in P dissolution. Therefore, HNP and MNP emerge as promising high-reactive P fertilizers for direct agricultural use and have a great potential as a source of P/Ca liquid fertilizer after nutrients balancing. So, eco-friendly grinding offers a potential approach to maximize PRs' agronomic potential, but long-term environmental impacts should be evaluated.

## 1. Introduction

One of the major challenges facing agriculture is the insufficient resource of phosphorus (P) in soils, which is an essential nutrient for enhancing soil fertility and crop

**Data availability statement:** All relevant data are within the paper and its Supporting Information files.

**Funding:** The authors extend their appreciation to the Deanship of Scientific Research at King Khalid University for funding this work through a large research project under grant number RGP 2/200/46. The authors also extend their appreciation to the Deanship of Scientific Research at Northern Border University, Arar, KSA, for funding this research work through project number "NBU-FFR-2025-1688-03.

**Competing interests:** The authors have declared that no competing interests exist.

production [1]. Phosphate rocks (Rs) are a vital source of all phosphorus (P) industries and play a critical role in feeding the world's growing population [2]. Additionally, they serve as the primary source of phosphorus-based chemicals [3,4].

However, a significant portion of soluble P in soil is rapidly converted into forms that reduce its availability to plants [5]. As a result, repeated applications of P compounds may be crucial throughout a crop's life cycle to maintain adequate levels of plant nutrition. Over time, excessive use of soluble P fertilizers can lead to soil saturation and and/or increase in the amount of immobile P and its waste, potentially resulting in increased P losses [6].

The phosphatic ore-based industries have an indirect hazardous environmental impact that can be seen in the increased occurrence of eutrophication. This is correlated with the extreme use of fertilizers for agricultural purposes, which leads to soil and water pollution [7]. The soluble phosphorus that exceeds plant requirements is either discharged into aquatic systems, resulting in algal blooms that can block sunlight and cause oxygen depletion, leading to toxicity and mortality of organisms, or form aluminum-phosphate that is insoluble. Consequently, it is crucial to find methods that not only increase soil phosphorus content for agricultural production but also reduce environmental pollution for ecological preservation [8].

Phosphate resources, which are categorized into four types based on their origin: 1) sedimentary marine sediments; 2) igneous deposits;3) metamorphic deposits; 4) biogenic deposits (bird and bat guano accumulations), are widely distributed in many parts of the world [9–11]. Most of the phosphate production comes from sedimentary sediments [12].

Egypt is one of the major countries having high reserves of sedimentary phosphate of Late Cretaceous age, with three main districts: Red Sea, Nile Valley and Abu-Tartur deposits. These deposits were formed in a reducing environment and have a grey-to-black colour due to organic matters enrichment. Over time, geochemical weathering has partially changed its colour to yellowish-brown due to oxidation by the experienced diagenetic processes [13,14].

Phosphate rocks, PRs, cannot be employed as fertilizers in their pristine form; their low solubility requires prior processing. This is correlated with their associated gangue minerals such as carbonates, silicates, feldspar, mica, carbonate minerals, and clays that require processing [15–17]. These processes include screening, scrubbing, heavy media separation, washing, roasting, calcination, leaching, and flotation [16,18]. The processing of phosphate ore fractions is significantly influenced by their particle size and had a positive effect on P solubility [19].

To address the ever-growing demands for phosphatic fertilizers, this study aims to develop new types of nonconventional and sustainable fertilizers that could improve fertilizer efficiency and performance through the application of new emerging nano-fertilizers technology. These nano-fertilizers, which can come in powder or liquid forms provide readily available nutrients to plants that can enhance plant production compared to traditional fertilizers.

## 2. Materials and methods

### 2.1 Materials and chemicals

Oxidized phosphate rocks from the different exposed grades (high, medium, and low) of Duwi Formation at Abu Tartur plateau, were used in the current study at Abu Tartur plateau after taking the required permissions from Phosphate Misr company. No additional permits were required for sample collection. Where permissions were granted verbally. The investigated samples were collected from the eastern to the western part of Liffiyia – Maghrabi sector (25°25′34″N and 30°05′08″E) that located about 50 km west of Kharga Oasis, Western Desert, Egypt (Fig 1). To avoid the malignant impact of inorganic acids (e.g., sulfuric, nitric and hydrochloric acid) although their high leaching capability, organic acetic acid (98%) as one of the most promising leaching agents for calcareous and/or dolomitic gangue materials associated with the phosphatic deposits was applied in the current study. Also, distilled water was used as diluting solution.

### 2.2 Samples preparation

To prepare the collected oxidized phosphate rocks (PR), the samples of each grade were crushed using a jaw crusher to get a size fraction of ‹ 4.75 mm. Then the crushed samples were delivered to the disc mill to get finer sizes of ≥ 100 μm.

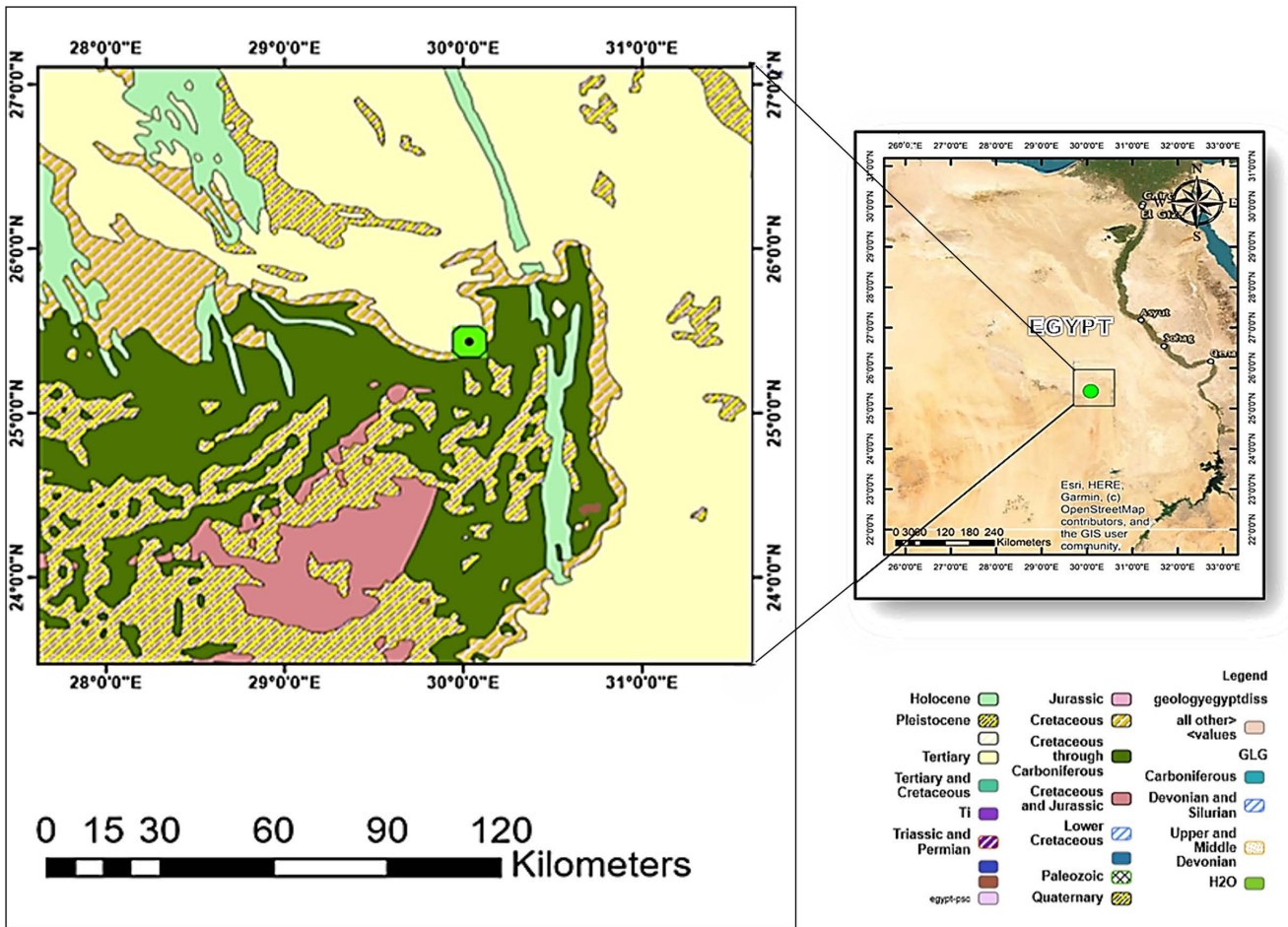

**Fig 1. Landsat and geologic map images.** These images show the location and geology of the study area at Abu-Tartur plateau, Western Desert Egypt.

The ground samples were mixed carefully till the entire homogenization. Then, the produced homogenous mixture was quartered several times to get a representative sample. After washing several times with distilled water and drying in an oven drier at 70˚C overnight, the sample was smoothly reground in an agate mortar and sieved to get a size < 100 μm. Each one of these ground samples was divided into two halves, one half was packed and labeled for further investigations. Whereas the other one was subjected to intensive grinding several times (sieving and grinding) using a ball mill (RETSCH PM 100/Germany) to get size fractions in nano-size (<100nm). The nano-size fractions were labeled as HNP, MNP and LNP, reflecting the high, medium and low grades of phosphatic deposits, respectively. Whereas those prepared fractions in micro-sizes were labeled as HMP, MMP and LMP, orderly.

## 2.3 Characterization of the prepared samples

The prepared PR samples in both sizes (micro-and nano-sizes) were characterized using the X-ray fluorescence (XRF) technique (Philips X-ray fluorescence analyzer model PW/2404) to determine their chemical composition, especially the $P_2O_5$ and CaO contents. Additionally, the mineralogical characteristics were examined using both optical microscopy (Nickon Eclipse: LV 100POL) and X-ray diffraction (XRD) techniques, Philips APD-3720 diffractometer (Cu Kα radiation operated at 20 mA and 40 kV) in the 2θ range of 5°-80° at a scanning speed of 5°/ min. Even though the morphological characteristics and the dominating functional groups on the surface of the addressed samples were investigated using the scanning electron microscopy (SEM) technique (JSM-6700F, JEOL, Tokyo, Japan, beam energy: 20–30 kV, working distance:11.1–12.2 mm) and Fourier transforms infrared (FT-IR) technique (Bruker FTIR-2000 Spectrometer with the mode of reflection at a 4 cm$^{-1}$ resolution), respectively. For SEM analysis, the powder of each sample was installed on stubs and covered with gold using a coating device (JEOL-JSM-420, Japan). Whereas, for the FT-IR study, the sample powder was mixed with KBr (Merck) in a 1:10 ratio to get translucent pellets used for identifying the functional groups of the samples. Furthermore, the Beta surface area ($S_{BET}$, m$^2$/ g), total pore volume ($V_t$, cm$^3$/ g), and average pore diameter (Dp, nm) of all the PR fractions (HMP, HNP, MMP, MNP, LMP, and LNP) were studied by Surface Area Analyzer (Nova 2000 Quantachrome) in nitrogen adsorption/desorption environment after degassing at 100°C/2h to remove the adsorbed contaminants from their surfaces and pores. The BET surface area ($S_{BET}$) was estimated by Brunauer-Emmet and Teller model [20], while the total pore volume (Vt) and average pore diameter (Dp) were measured using Barrett-Joyner-Halenda (BJH) formula [21].

## 2.4 Leaching experiments using acetic acid

To evaluate the solubility of the prepared samples (HMP, HNP, MMP, MNP, LMP, and LNP) of the investigated phosphate deposits by liquified acetic acid, AA (98%) with specified volume of distilled water (25 ml), the following experimental parameters were studied at ambient temperature: effect of AA concentration, effect of contact time and regeneration studies. The prevailing experimental conditions of each experimental parameter are compiled in Table 1. After solid-liquid

**Table 1. Applied experimental parameters.** This table elaborates the prevailing conditions during the phosphorus (P) leaching experiments from the prepared phosphatic rocks (PRs) by acetic acid (AA).

| Investigated parameter | Prevailing Conditions | | | | | The other parameters |
|---|---|---|---|---|---|---|
| AA/PR ratio (w/w) | (0.5:1) 0.75: 1.5g | (1:1) 1.5:1.5g | (2:1) 3:1.5g | (3:1) 4.5:1.5g | (4:1) 6:1.5g | 25 ml distilled water, 200 rpm/2h (speed/agitation time). |
| Contact time (h) | ½ | 1 | 2 | 4 | 6 | AA/PR (2:1 ratio, 3g AA: 1.5g PR), 25 ml distilled water, 200 rpm (speed) |
| Regeneration studies of PRs (g) | Cycle 1 (10:5g) | Cycle 2 (6:3g) | Cycle 3 (4:2g) | Cycle 4 (3:1.5g) | Cycle 5 (1:0.5g) | AA/PR (2:1) ratio, 25 distilled water, 200 rpm/2h (speed/agitation time). |

separation using digital centrifuge (Frontier <u>TH5706/Ohaus/USA</u>) at 8000 rpm/10 min, the liquidous phases were analyzed via inductive coupled plasma optical emission spectrometry (ICP OES/Optima DV) to estimate the dissolved P obtained from the leaching process. For regeneration studies, a 2:1 w/w ratio of AA/PR for each of the prepared samples were used separately and the dissolved P was estimated after centrifuging. Then the solid filtrate fractions were collected and dried overnight at 105 °C to be reused again for another cycle of acid leaching. These fractions were reweighted again before the reconduction of the leaching process to estimate the wasted mass from the previous cycle and to maintain the AA/PA ratio at 2:1 in each new cycle. This step was repeated several times till the approximate consumption of the applied powder of the PR (<u>Table 1</u>).

## 3. Results and discussion

### 3.1 Field characteristics

The phosphatic rocks (PRs) of the current study that were collected from Abu-Tartur plateau, belong to Duwi Formation of Campanian–Maastrichtian age (<u>Fig 2a</u> and <u>2b</u>). Such deposits represent one of the largest phosphate reserves in the world, about 70 billion metric tons with different grades up to 30 wt.% $P_2O_5$ [22]. Also these phosphatic deposits exhibit lateral and vertical facies variation, as well as thickness reduction towards both east and west directions of the sector [23]. The primary or secondary origin of these deposits till now is a matter of debate among scientists [24]. Duwi Formation composed of three dominant members: Abu Tartur Phosphate (lower), Liffiya Shale (middle) and Maghrabi Shale (upper) [23]. The bottom of the lower member is composed mainly of 6 m-thick PR, whereas the top part is up to 5 m-thick of dark grey shale and siltstones intercalations [23]. Such member was considered as the initial marine transgression record for the Late Cretaceous of Egypt that persisted till the deposition of organic matters/pyrite enriched shale [25,26]. On the other hand, the middle Liffiya Member is represented by 13 m-thick regressive alternations of light green, medium-grained glauconite-bearing mudstone and siltstones, calcareous siltstone and phosphate. Also, the upper Maghrabi Member (12 m-thick) is composed of a fresh transgression greyish shale with glauconite bed intercalation that is overlain by a shallow marine, regressive, thin phosphorite bed and siliciclastic gravel [24].

### 3.2 Characterization of prepared samples

The petrographical characteristics of the pristine PRs for the micro-fractions (HMP, MMP and LMP) were examined, using transmitted light optical microscope at different magnifications (<u>Fig 3a</u>–<u>3i</u>). The detailed microscopic investigation revealed

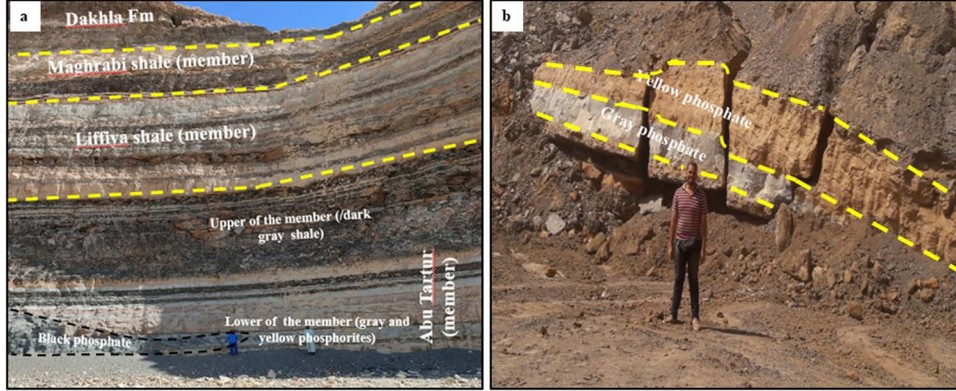

**Fig 2. Field photograph.** This photograph shows the stratigraphy of Duwi Formation: (a) Eastern part, (b) Western part of Liffyia - Maghrabi Sector in Abu Tartur mine.

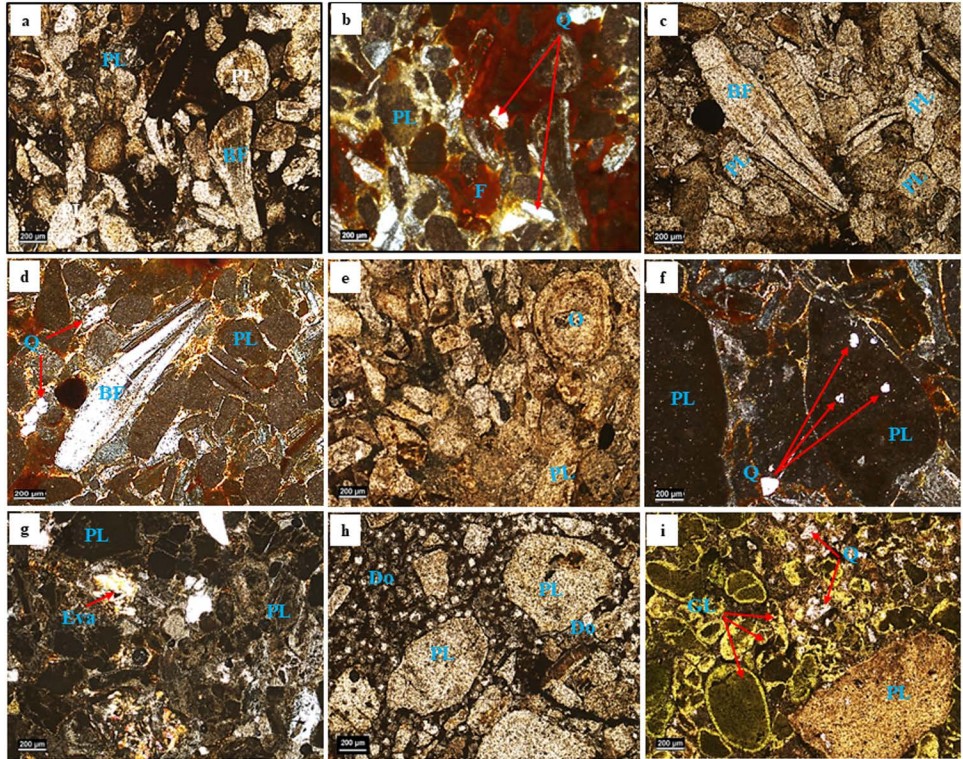

**Fig 3. Petrographical images.** These images display the high-grade phosphate (HMP) in PPL (a, c and e) and XPL visions (b, d and f): Peloid grains (PL), Dolomite rhombs (D), Ooids (O), Fish bones (FB), Quartz (Q) and Iron oxide (F); Medium-grade phosphate (MMP) in PPL (g), and low-grade phosphates in XPL visions (h and i): Peloid grains (PL), Evaporites (Eva), Dolomite rhombs (D), Quartz (Q) and Glauconite (G).

that the dominant composition of all the addressed PR grades are mudclasts (peloids: fragments of phosphatic mudstones, but sometimes contain silt-sized detrital grains such as quartz [27] and bioclasts (bone fragments and teeth) [27] with marked privilege of the former in the high-grade PRs compared to the other grades (Fig 3). While the non-phosphatic gangue minerals are represented by carbonates (dolomite, calcite and ankerite), quartz, pyrite and evaporites (gypsum and anhydrite), aligning with previous data [28]. Mudclasts are generally presented as rounded to sub-rounded homogeneous pellets embedded in calcareous/ferruginous cement. These pellets are predominantly structureless except for the occasionally occurring ooids that formed by the growth and crystallization of micrometric phosphate layers (Fig 3e). As well, mudclasts generally display different shade of colors ranging from white to brown in PPL that become isotropic between XPL (Fig 3a and 3b), agreeing with other investigations [29]. However, black varieties can be confined to fresh PR samples. On the other hand, phosphatic bioclasts are represented by bone fragments and teeth (Fig 3a, 3c and 3d). Unlike the low-grade PR, bone fragments have special concentration in both high and medium grade PRs. They occur either as angular to subangular prismatic grains or as irregular ones. They are generally anisotropic with low birefringence, grey first-order interference color lamellar twinning and/or undulatory extinction (Fig 3d). Elongated and prismatic bone fragments occasionally show extinction parallel to their axis of elongation (Fig 3d) [30,31]. Sometimes, the core of some of these bone fragments are occupied with disseminated pyrite surrounded by a yellowish oxidized rim [29]. Similarly, the dominant gangue minerals in the high-grade PRs are presented as scattered detrital quartz (Fig 3f), whereas needle-shaped crystals and stellar arrangements of Ca-sulfate (gypsum and anhydrite) are the prevailing gangues in the medium-grade varieties (Fig 3g). On the other hand, the low-grade PRs are marked by high frequency of dolomite

rhombohedra surrounding the phosphatic grains, are predominantly composed of minerals like fluorapatite ($Ca_5(PO_4)_3F$) and francolite ($Ca_5(PO_4,CO_3)_3F$), but the most commonly encountered variant is francolite (Fig 3b and 3c).

The chemical composition of the investigated PRs in both micro- and nano-fractions (HMP, HNP, MMP, MNP, LMP and LNP) were obtained by XRF technique as given in Table 2. The classification of these fractions into high, medium and low-grade PRs is based on their $P_2O_5$ contents: 35.7, 27. 92 and 8.01 wt. % in HMP, MMP and LMP, respectively (Table 2). Such $P_2O_5$ contents can be considered as a merit of their phosphatic components (bone-and mud-fragments) as discussed in the petrographical investigations. Likewise, the distinct variations in many of the other major oxides of these fractions were observed, reflecting a drastic difference in their mineralogical composition, particularly their non-phosphatic components. Concerning the PRs micro-fractions, LMP revealed higher contents of $Fe_2O_3$, $SiO_2$, MgO, $Al_2O_3$, MnO and LOI (7.03, 5.96, 9.65, 1.39, 1.22 and 32.27 wt. %, orderly) compared to their counterparts of the other grade fractions: high, HMP, and medium, MMP (Table 2). The high $SiO_2$ content in LMP can be ascribed to the abundant presence of glauconite and some detrital quartz in this fraction matching with the results of the petrographical investigations. Whereas the approximately high $SiO_2$ content (4.91 wt.%) in HMP can be correlated to the presence of detrital quartz in this fraction (Fig 3f). Similarly, the high MgO, MnO and LOI contents in the LMP (Table 2) can be attributed to the intensive presence of ankerite/dolomite and glauconite as was displayed by petrographical investigations.

As well, the remarkable abundance of $SO_3$ (15.03 wt.%) in the MMP in comparison with the other studied micro-fractions can be correlated to the intensive presence of gypsum/ anhydrite, orderly, on expense of oxidized pyrite via experienced diagenetic processes, aligning with petrographical outcomes and previously reported data [27,32]. Other oxides (e.g., $TiO_2$ and $Na_2O$) occur in all the investigated PR grades in micro-sizes with no distinct variations in agreement with previously reported data [28]. Furthermore, the high depletion of fluorine content (F = 0.66 wt.%) in the LMP compared to the other comparable grade fractions, not only correlated to its low phosphatic component content but also due to the re-crystallization processes of francolite into brushite [$CaPO_3(OH)\cdot 2H_2O$] that accompanied the experienced long and lasting diagenetic phases [33,34].

Finally, CaO followed the same trend as the $P_2O_5$ in the studied micro-fractions with respect to the phosphatic components in each of them (Table 2).

After intensive grinding to nanoscale, both $Fe_2O_3$ and LOI contents in HNP and MNP increased (Table 2). The rise in $Fe_2O_3$ content can be attributed to the oxidation of ferrous components, such as $Fe^{2+}$-bearing minerals like glauconite,

**Table 2. XRF results. This table illustrates the chemical composition of the prepared PRs: HMP, HNP, MMP, MNP, LMP and LNP.**

| Major elements | HMP | HNP | MMP | MNP | LMP | LNP |
|---|---|---|---|---|---|---|
| $SiO_2$ | 4.91 | 6.54 | 3.09 | 3.46 | 5.96 | 5.47 |
| $TiO_2$ | 0.05 | 0.07 | 0.04 | 0.04 | 0.05 | 0.06 |
| $Al_2O_3$ | 1.10 | 1.22 | 0.67 | 0.65 | 1.39 | 1.04 |
| $Fe_2O_3$ | 3.20 | 4.79 | 5.08 | 5.56 | 7.03 | 7.35 |
| MnO | 0.24 | 0.26 | 0.11 | 0.12 | 1.22 | 0.89 |
| MgO | 0.30 | 0.33 | 0.18 | 0.16 | 9.65 | 9.64 |
| CaO | 41.40 | 38.43 | 34.19 | 33.24 | 29.48 | 28.71 |
| $Na_2O$ | 0.61 | 0.53 | 0.44 | 0.42 | 0.18 | 0.22 |
| $K_2O$ | 0.08 | 0.10 | 0.20 | 0.15 | 0.09 | 0.08 |
| $P_2O_5$ | 35.70 | 31.23 | 27.92 | 24.22 | 8.01 | 8.90 |
| $SO_3$ | 3.69 | 3.54 | 15.03 | 16.90 | 3.83 | 4.27 |
| F | 2.18 | 1.70 | 1.61 | 1.51 | 0.66 | 0.56 |
| CL | 0.07 | 0.08 | 0.08 | 0.08 | 0.04 | 0.06 |
| LOI | 6.04 | 10.62 | 10.92 | 13.00 | 32.27 | 32.64 |

ankerite, and pyrite, which were present in the precursor micro-phosphate fractions before grinding [35]. Similarly, the elevated LOI values suggest greater moisture retention, likely due to structural disorder and increased surface energy in both phosphatic and non-phosphatic components because of intensive grinding. However, interpreting the impact of crystalline structure disruption remains complex, particularly regarding peak broadening and the quantification of amorphous content [35,36],Conversely, the increase in $Fe_2O_3$ and LOI of the LNP sample was insignificant in comparison with the precursor LMP and the other nano-fractions. This could be attributed to the experienced complexity in the grinding process of its precursor LMP to nano-fraction because of the high LOI content that exceeded 32 wt.% (Table 2).

Moreover, the $P_2O_5$ content decreased in HNP and MNP, whereas it slightly increased in LNP from 8.01 to 8.9 wt.% (Table 2). This reduction in HNP and MNP can be attributed to the preferential breakdown of francolite (carbonate-fluorapatite) into smaller, more reactive fragments during intensive grinding. This process may have led to the release of loosely bound phosphate species, which could have been lost due to surface interactions or volatilization. Additionally, grinding-induced structural disorder may have influenced the redistribution of elements, impacting XRF quantification. In contrast, the slight increase in $P_2O_5$ content in LNP could be due to the mechanical breakdown of gangue minerals (e.g., dolomite, ankerite, or silicates), which exposed more phosphate-rich regions that were previously embedded within the mineral matrix. Similarly, the CaO and F contents were slightly reduced after grinding, especially in the high and medium grade fractions of nano-size (Table 2). This decrease is unlikely due to volatilization, as neither Ca nor F are volatile under the applied grinding conditions. Instead, it may be attributed to the mechanical breakdown of carbonate-fluorapatite (francolite), leading to the redistribution of Ca and F within the sample matrix rather than an actual loss. As well, the decomposition of Ca-Sulfate Phases (Gypsum and Anhydrite in MNP) may have affected the hydration state of gypsum ($CaSO_4 \cdot 2H_2O$), leading to partial transformation into anhydrite ($CaSO_4$), which could impact XRF detection of CaO. Additionally, the grinding-induced moderate increase in disorder (subtle broadening reflection increase) and particle agglomeration could have altered XRF detection efficiency, affecting the apparent concentrations. This effect is likely due to changes in sample homogeneity and particle packing rather than a true compositional depletion. Furthermore, the positive or negative variation of other oxides after grinding was not substantial.

The XRD patterns (Fig 4a–4c) of PR samples in both sizes (micro: HMP, MMP and LMP and nano-sizes: HNP, MNP and LNP) revealed that phosphatic and non-phosphatic minerals are the main components but in different contents, matching with the outcomes of petrographical investigations. The phosphatic minerals are represented by francolite (Carbonate-fluorapatite [$Ca_5$ ($PO_4$, $CO_3$)$_3$ (OH, F)]) and brushite (Calcium Phosphate Hydroxide Hydrate [$CaPO_3$ (OH) $\cdot 2H_2O$]), but with less frequency. The remarkable appearance of these apatite minerals at HMP compared to MMP and LMP elucidates the high $P_2O_5$ content in such fraction, aligning with XRF data. The minor content of brushite mineral in all the PR grades could be attributed to the recrystallization of phosphatic minerals during the experienced diagenetic processes [33]. Non-phosphatic minerals in the HMP are represented by quartz (2θ: 26.01° and 42.47°), pyrite (2θ: 28.23°), and hematite at 2θ: 64° (Fig 4a). Concerning MMP, sulfate minerals such as gypsum (2θ: 29.22°; 31.96° and 48.47°) and anhydrite (2θ: 25.93°) in addition to quartz (2θ: 40.23°) are the main non-phosphatic minerals (Fig 4b). Whereas ankerite (2θ: 24.29°, 31.24°, 41.40°, 43.93°, 45.18°, 51.3°, 59.1°, 60° and 67.67°) is the prevailing non-phosphatic mineral in LMP with appreciable presence of dolomite (2θ: 43.58°), glauconite (2θ: 26.95° and 37.65°), quartz (2θ: 26.17°) and gypsum at 2θ: 33.72° (Fig 4c). The domination of non-phosphatic minerals at the expense of the phosphatic ones explains the low $P_2O_5$ content of LMP in comparison with the other two grades as assured by the XRF data. After grinding to nano-sized fractions (HNP, MNP, and LNP) a subtle broadening of reflections was observed in the characterizing peaks of the phosphatic/ non-phosphatic minerals especially at HNP, indicating a moderate increase in structural disorder (Fig 4a) [35]. This effect suggests a reduction in crystallite size and a partial disruption of long-range order due to intensive grinding. However, the data does not indicate complete amorphization, as the primary diffraction peaks remain visible. So, the observed broadening in HNP, MNP samples is consistent with a transition toward increased disorder but still retains a degree of crystallinity [35]. On the contrary, the XRD patterns indicate that grinding had only a moderate impact on the crystallinity of

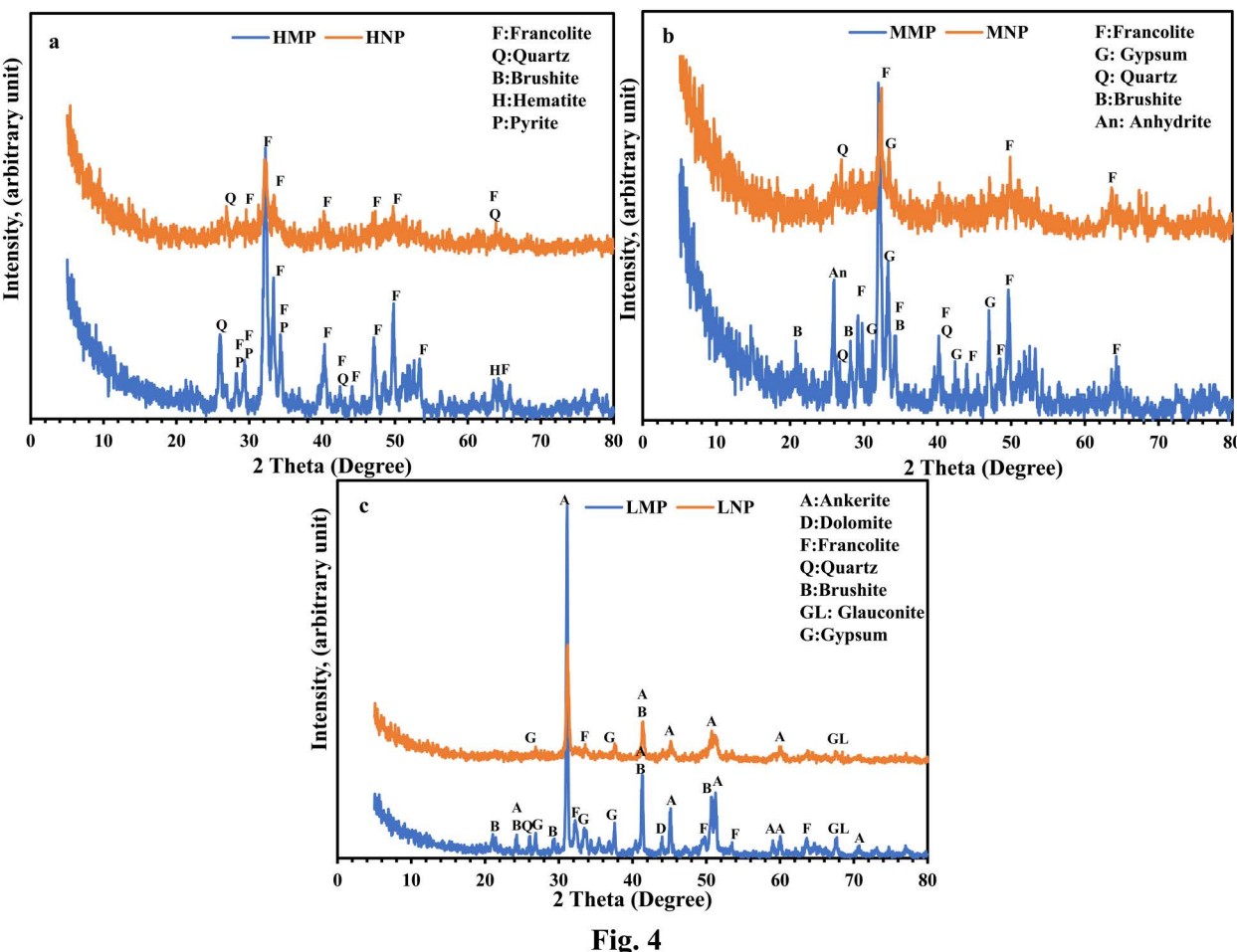

**Fig 4. XRD patterns.** These patterns display the precursor micro-size PR samples in comparison with its modified nano-sized samples: HMP/ HNP (a), MMP/ MNP (b) and (c) LMP/LNP.

LMP (Fig 4c), as evidenced by the slight broadening of reflections. This limited structural disruption could be attributed to the high LOI content (>32 wt.%), which likely hindered effective size reduction by promoting agglomeration and reducing grinding efficiency, as also suggested by XRF data.

The FT-IR spectra (Fig 5), displays the main functional groups of both mico- and nano-sizes fractions of the investigated PRs (HMP, HNP, MMP, MNP, LMP and LNP). The broad vibration bands around 3405 and 1630 cm$^{-1}$ in PR micro-fractions (HMP, MMP and LMP), can be ascribed to the stretching mode of hydroxyl group [11,37–40]. Also, the weak absorption bands at 1623–1632 cm$^{-1}$, were also assigned to the bending mode of water "δ ($H_2O$)" [41], that are typically associated with hydrous minerals of the investigated samples as glauconite and gypsum, matching with XRD and petrographical data [42]. The larger intensity of hydroxyl groups in both stretching and bending modes in LMP compared to other fractions (HMP and MMP) can be attributed to its significant LOI content and the remarkable presence of hydrous minerals (glauconite and gypsum) in such fractions as elaborated by XRF, XRD and petrographical investigations, respectively. As well, absorption bands at 1428.9, 1428.7 and 1445.6 cm$^{-1}$, for HMP, MMP and LMP, orderly, can be correlated to the anti-symmetrical elastic vibration (v$_3$) of carbonate group ($CO_3$)$^{2-}$ [40,43,44]. Whereas the v$_2$ elastic vibration of carbonate ($CO_3$)$^{2-}$ group was shown only in LMP around 879 cm$^{-1}$ [45]. The conspicuous occurrence of ($CO_3$)$^{2-}$ group in

both vibration types in LMP can be correlated to superior attendance of ankerite and dolomite minerals in such fractions as compared to HMP and MMP. On the other hand, the anti-symmetrical elastic vibration ($v_3$) bands at 1045.6, 1046 and 1041.6 cm$^{-1}$ for HMP, MMP and LMP, orderly, can be assigned to the stretching vibrations of Si-O bond of the inorganic components as quartz and glauconite, aligning with XRD and petrographical data, with some preference of HMP on the level of intensity [46]. Additionally, the changing in the vibration bands that reflect the presence of phosphate group $(PO_4)^{3-}$ with $v_4$ absorption pattern, was observed at 604.3, 605 and 590 cm$^{-1}$ in HMP, MMP and LMP micro-size PR fractions, orderly [45].

The grinding process to nano-sizes (HNP, MNP and LNP) exerted either positive or negative shifts in the frequencies of the main functional groups of the micro-sizes fractions accompanied by an obvious strengthening in the intensity of some encountered groups such as OH, δ ($H_2O$), Si-O, $(CO_3)^{2-}$ and $(PO_4)^{3-}$, confirming the successful impact of such process. For example, the bands that were ascribed to $v_2$ and $v_3$ elastic vibration of $(CO_3)^{2-}$ group, were remarkably strengthen in LNP although crystallinity reduction of carbonate minerals (ankerite and dolomite). This could be correlated with the probable formation of carbonic acid ($H_2CO_3$) from the released carbon dioxide ($CO_2$) and water vapor ($H_2O$) [47,42]; the release of these volatiles in an approximately closed grinding system has triggered pressure building up, justifying the remerging of these vapors in $H_2CO_3$ form [48,49]. The latter probably further dissociate into bicarbonate ions ($HCO_3^-$) that could combine with other cations released from the decomposition of other minerals such as potassium ($K^+$) from glauconite or $Ca^{2+}$ from sulphate/phosphate minerals to form amorphous bicarbonate, explaining the intensification of the absorption bands of $(CO_3)^{2-}$ group in LNP [50]. Similarly, the strength of Si-O and $(PO_4)^{3-}$ groups in the nano-size fractions could be attributed to the reduction in crystallite size and increased structural disorder (i.e., subtle broadening of reflections rather than a significant loss of crystallinity) that was induced by intensive grinding, especially in the HNP and MNP (Fig 5). Conversely, both modes of water (stretching and bending) were reduced after grinding in LNP (Fig 5). Whereas in HNP and MNP, the absorption bands of these groups were slightly strengthened with grinding [51,52]. This could be attributed to the enlarged affinity to absorb moisture from the surrounding environment that accompanied the intensive reduction in particle sizes of these fractions, aligning with the LOI contents (Table 2).

The SEM images revealed the particle size distribution and the morphology of PR samples in both sizes micro: (HMP, MMP and LMP) and nano-sizes: (HNP, MNP and LNP) at different magnifications (Fig 6a–6l). The micro-size fractions (HMP, MMP and LMP) revealed heterogeneity in particle shapes and sizes in accordance with the difference in toughness and hardness of their mineral assemblages, especially the LMP, as assured by XRD and petrographical investigations. This heterogeneity could also be attributed to the differences in the LOI contents among these fractions in consequence of heat during the milling process as approved by XRF data (Table 2). Therefore, the HMP and to a lesser extent MMP revealed larger ratios of tinier particles than their counterpart LMP (Fig 6a, 6b, 6e, 6f, 6i and 6j, orderly). Generally, the common feature in all the addressed micro-size fractions (HMP, MMP and LMP) is the approximate sub-angularity of their particles with occasional appearance of flouc-like structures toward the rim of their agglomerated particles (Fig 6a, 6b, 6e, 6f, 6i and 6j, respectively). But with intensive grinding to nano-sizes (HNP, MNP and LNP), these approximately angular particles converted to spheroidal ones through rigorous rubbing against each other, as well as be the action of the grinding balls of various sizes (Fig 6c, 6d, 6g, 6h, 6k and 6l, orderly). However, the high LOI ratios contribute to a pervasive complication for the grinding process and hence the progressive agglomeration phenomena for the produced nanoparticles [53,54] as displayed in Fig 6k and 6l. Consequently, the particle-size distribution tends to be uniform at the high-grade nano-fraction (HNP) and to less extent at MNP in comparison with the LNP. This could be taken as a sign for the approximate destruction of the structure of the incorporated minerals (phosphatic and non-phosphatic ones) in these fractions (HMP and MMP) as displayed by the XRD results.

The $N_2$ adsorption-desorption isotherms of the analyzed PRs (HMP, HNP, MMP, MNP, LMP and LNP) are presented in Fig 7a–7c). These samples revealed type III adsorption-desorption isotherms in correlation with the weak adsorbent-adsorbate interaction [55]. Meanwhile, the displayed hysteresis loops, $H_3$- type of the IUPAC classification, assure the

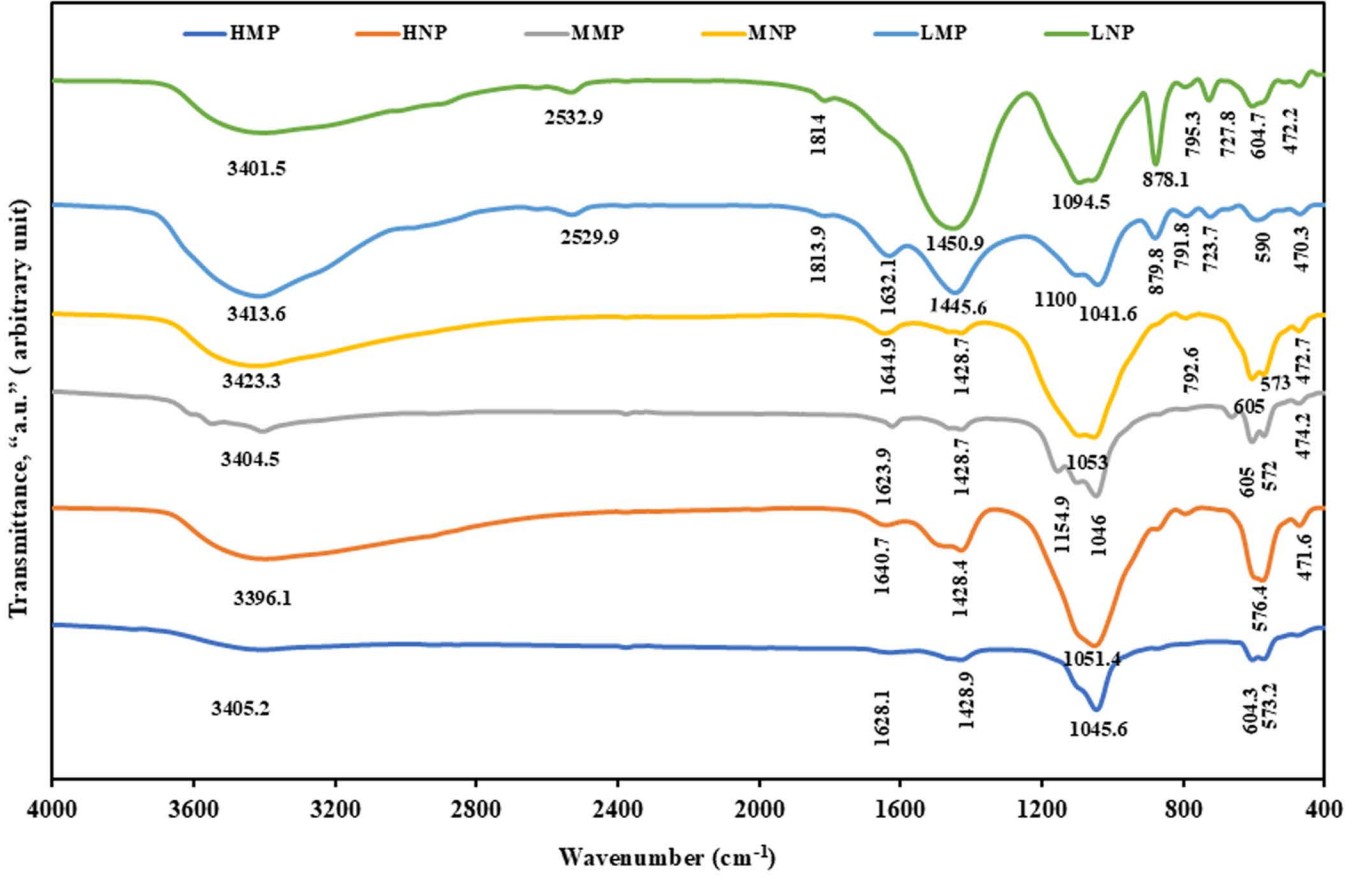

**Fig 5. FT-IR spectra.** These spectra display the precursor micro-size PR samples (HMP, MMP and LMP) in comparison with its modified nano-sized samples, HNP, MNP and LNP.

accommodation of these materials by slit-like pores [55,56]. Furthermore, these isotherms indicate mono/multi-layers adsorption of nitrogen molecules [57,58]. As well, all the presented isotherms, did not exhibit plateau stage (i.e., no equilibrium were attained) at high-pressure (P/P⁰ > 0.9), suggesting the presence of open micro-porosity [56,59]. Moreover, the rapid increase in adsorption branches of these isotherms, could be attributed to the capillary condensation [60,61]. On the other hand, the intermittent overlap of the isotherm branches at low P/P⁰, replicates fragile connectivity among the semi-closed pores of the investigated materials (Fig 7a–7c). But at higher P/P⁰, the divergence of these branches, especially in the micro fractions (HMP, MMP and LMP) over the nano-ones (HNP, MNP and LNP), implies the existence of open large pores in juxtaposition to semi-closed ones [62].

In agreement with $N_2$ isotherm results, the impact of intensive grinding upon the geometrical parameters ($S_{BET}$, $V_t$ and $D_p$) of the micro-size PRs, is shown in Table 3. This was elaborated by the remarkable reduction in $S_{BET}$ and $V_t$ of the produced nano-size varieties, especially HNP fraction (17.725 m²/g and 0.053 c³/g, respectively). Such a reduction could be tied to the partial demolition of the original crystalline structure of the produced nano- particles as was confirmed by XRD data (Fig 4a). These tiny particles tend to combine and pack together to form larger non-porous agglomerated aggregates by continual compression during the grinding process [63–65] as shown by SEM images (Fig 6c and 6d). This agglomeration phenomena could be interrelated with the liberation of the tightly bounded crystalline water with intensive grinding [66], with noticeable privilege for HNP over MNP and LNP fractions in descending order (Table 2). On the contrary, the

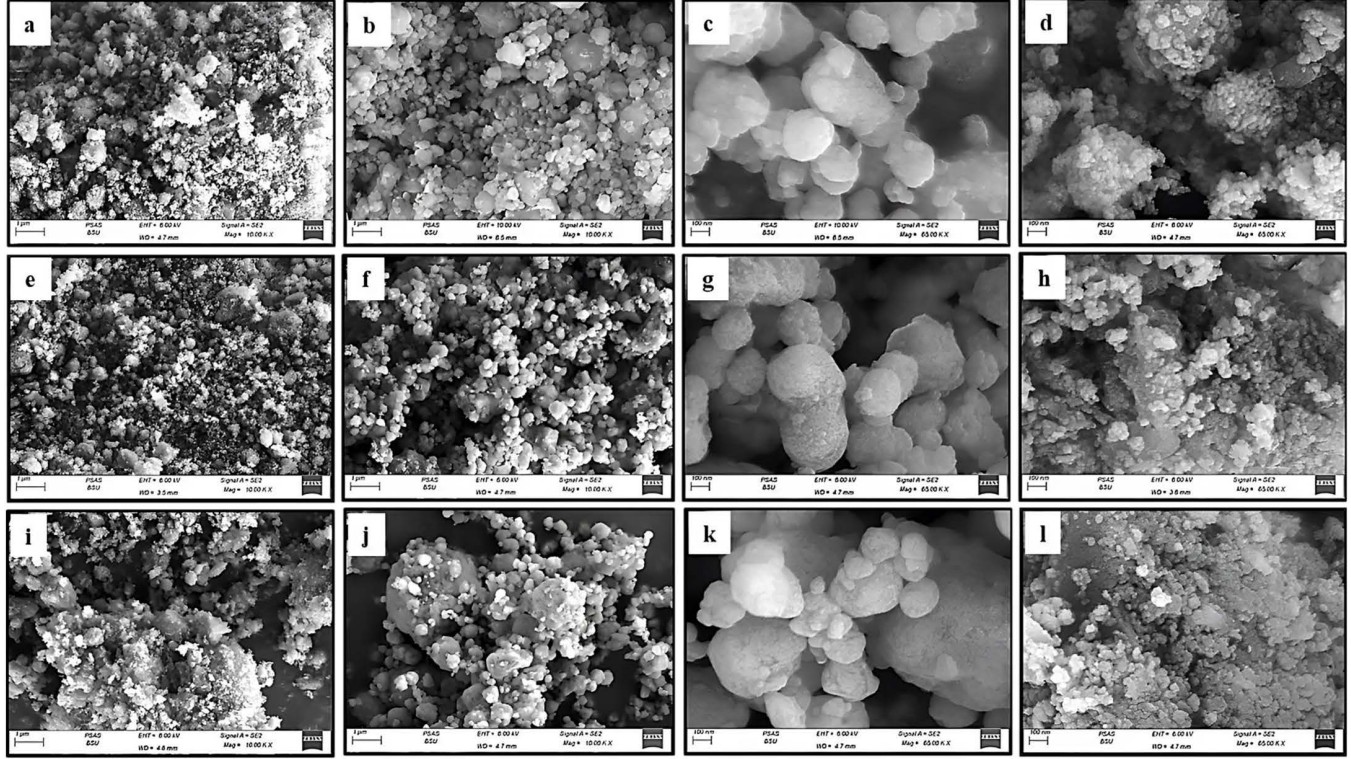

**Fig 6. SEM images.** These are the images of HMP (a and b), HNP (c and d), MMP (e and f), MNP (g and h), LMP (i and j) and LNP (k and l).

limited reduction in $S_{BET}$ (20.79 and $V_t$ of the LNP (20.79 m²/g and 0.041 c³/g, respectively) compared to its counterpart fraction (LMP = 29.14 m²/g and 0.087 c³/g, orderly), could be attributed to the hydrous nature of the majority of the crystalline phases of the latter (francolite, brushite, gypsum and glauconite) and the domination of carbonate phases (Ankerite and dolomite) as was assured by both XRD and LOI (32.72 wt.%) shown in XRF results (Table 2). This composition reduced the impact of grinding upon LMP fraction and approximately hampered the complete demolition of the internal structure of the composing phases as was confirmed by XRD data (Fig 4c).

### 3.3 Leaching experiments

**3.3.1 Effect of acetic acid concentration.** The concentration of acetic acid (AA) had a profound influence on phosphorous, P, dissolution from the investigated PR samples (micro: HMP, MMP and LMP and nano-sizes: HNP, MNP and LNP) using variable AA/PR ratios (w/w) that ranged from 0.5:1–4:1. To elaborate this influence, the leaching experiments were executed for each sample separately at a constant PR mass (1.5 g), 200 rpm/2h and 25 ml of distilled water as a liquid medium at room temperature (Fig 8a). For all the investigated samples, it was revealed that the P dissolution rate was progressively improved with increasing the applied acid ratio from 0.5:1–2:1. Beyond this ratio, the improvement in P dissolution rate was irrelevant, indicating equilibrium state attainment [67]. This aligns with the fact that P solubility increase with increasing the medium acidity [68–70]. In such acidic medium, the protonation of the ≡P–O⁻ surface group of the phosphatic minerals (francolite and brushite) by the H⁺ protons of the employed acetic acid, act as a sole driving force for P dissolution [71]. However, intervening from the organic ligand (mono-carboxylic group) of the acetic acid upon P dissolution was negligible as it lacks donor atoms (two or more), like in the other organic acids (oxalic and citric acids), capable of forming bi- or multi-dentate mononuclear surface chelates, and that could accelerate cleavage

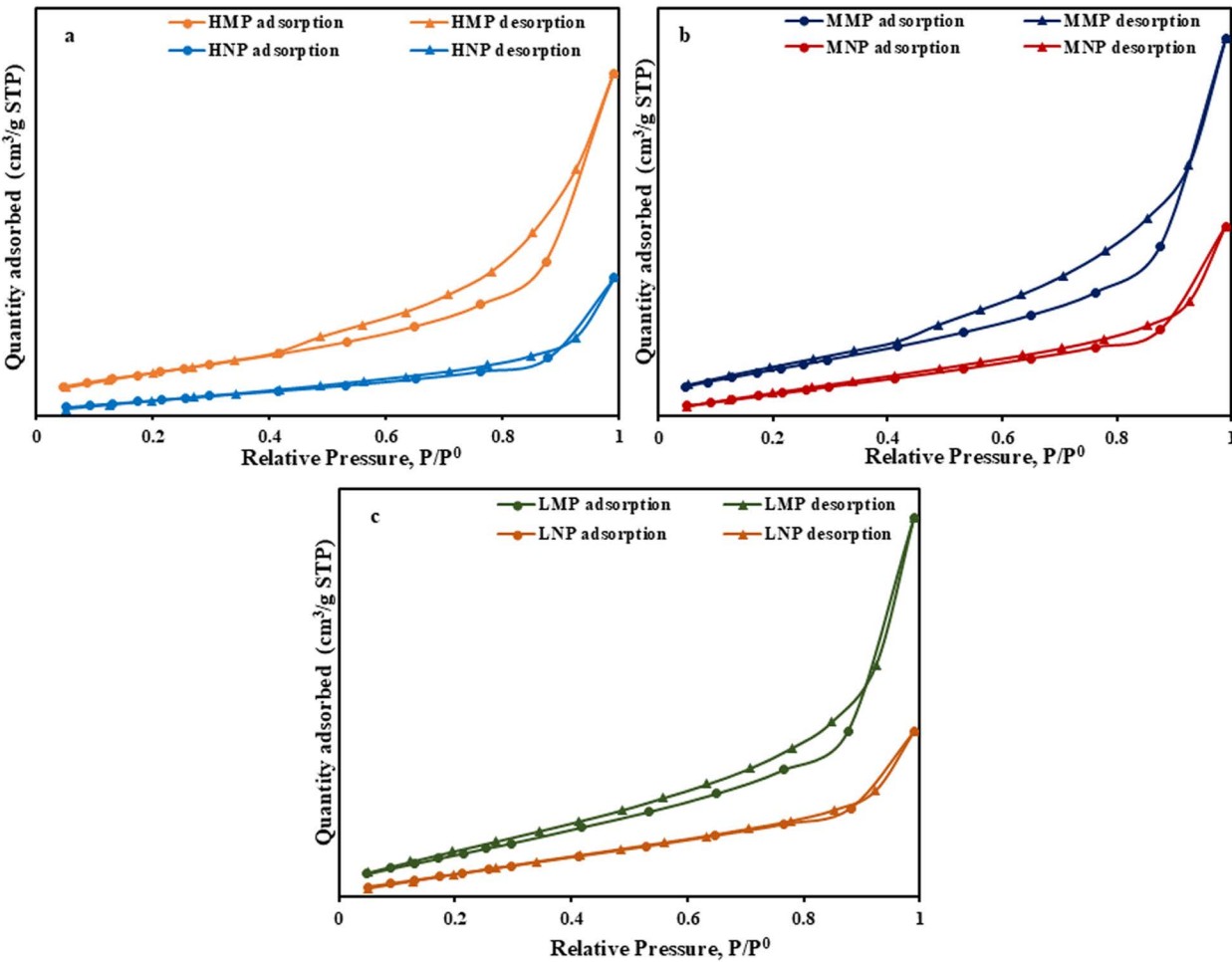

**Fig 7. N$_2$ adsorption-desorption isotherms.** These are the isotherms of HMP/HNP (a), MMP/MNP (b) and LMP/LNP (c).

of metal-oxygen bonds to be eventual detached from mineral surfaces [72]. Additionally, it was revealed that the P dissolution rate from both micro- and nano- PR fractions followed the following trend: HNP (622–1994ppm)> MNP (555–1641ppm)>HMP (173–499ppm)> MMP (165–468ppm)>LNP (24–176ppm)> LMP (3–65ppm). This trend was not only correlated with the P$_2$O$_5$ wt. % contents of these investigated fractions, but also on their particle sizes, and crystallinity/

**Table 3. Textural parameters.** This table displays the textural parameters of micro- and nano-size PR fractions obtained from N2 adsorption-desorption isotherms.

| Sample | Surface Area (m²/g) | Total pore volume (cm³/g) | Average pore diameter (nm) |
|---|---|---|---|
| | (S$_{BET}$) | V$_t$ | D$_p$ |
| HMP | 42.27 | 0.13 | 1.94 |
| HNP | 17.73 | 0.05 | 1.92 |
| MMP | 42.01 | 0.13 | 1.93 |
| MNP | 26.21 | 0.07 | 1.92 |
| LMP | 29.14 | 0.09 | 1.93 |
| LNP | 20.79 | 0.04 | 1.91 |

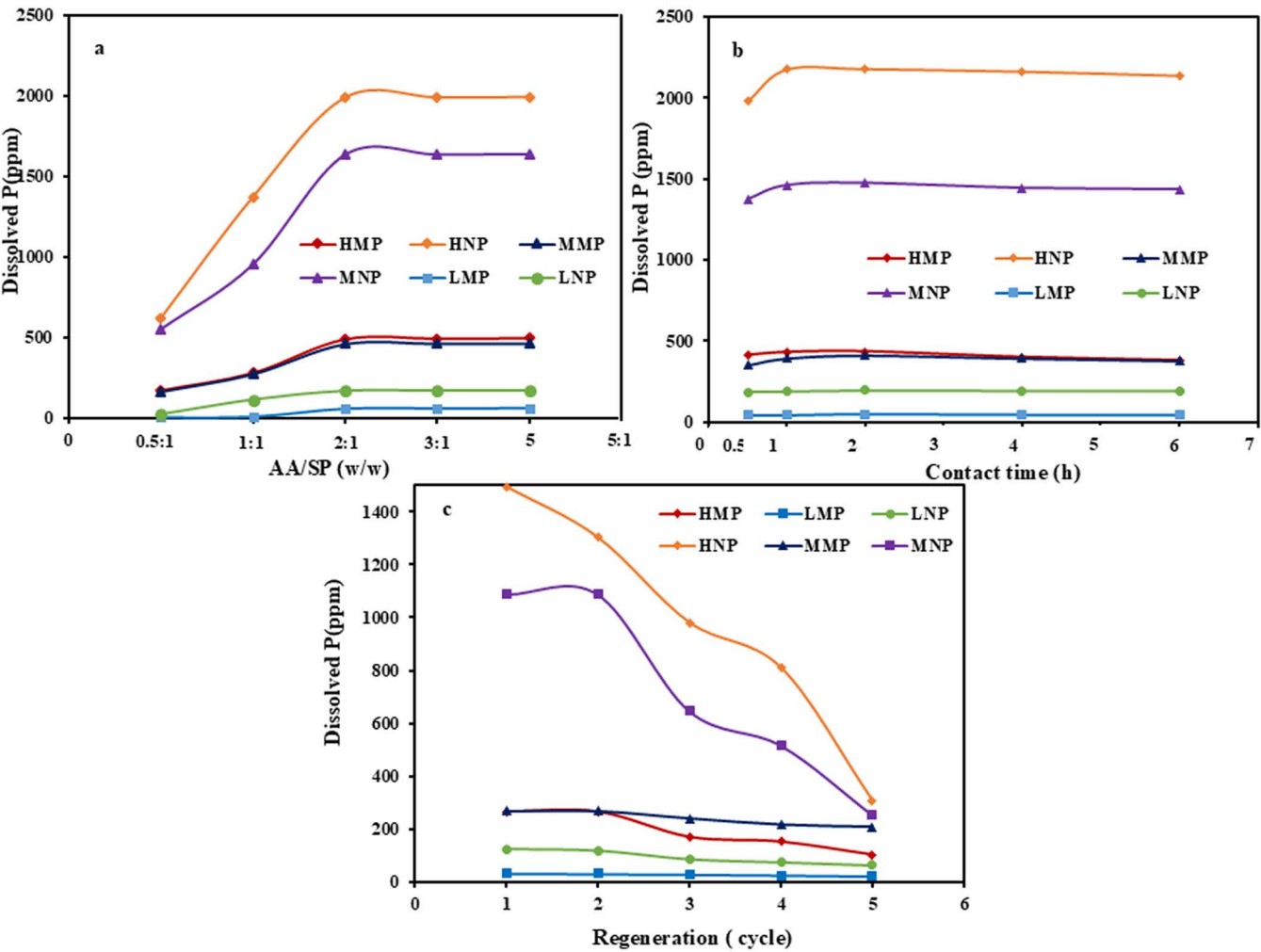

**Fig 8. Acetic acid (AA) leaching studies.** These studies highlight the effects of AA concentration (a) and reaction time (b) on P dissolution rate of both micro- and nanosized PR fractions, as well as the magnitude of P dissolution rate along five leaching cycles of both micro- and nanosized PR fractions (c) during reusability studies.

structural disorder nature. But despite the reduction in the $P_2O_5$ wt. % contents of HNP and MNP, the prevalence of nanoparticles raised the solid/liquid interface, and hence P solubility was improved [19,73,74]. Similarly, the gained structural disorder/ subtle broadening of reflections of the phosphatic minerals with intensive grinding in the nano-fractions (especially HNP and MNP) improved their reactivity for acetic acid and hence their P dissolution rate, compared with their crystalline counterparts in the micro-ones, aligning with XRD data. On the contrary, the very low $P_2O_5$ wt. % content of both LNP and LMP made them occupy such late order, with preference of the former as was demonstrated by XRF data (Table 2). On the light of the current results, the 2:1 ratio of AA/PR was selected for the conduction of the subsequent experiments.

   **3.3.2 Effect of reaction time.** The influence of reaction time (0.5, 1, 2, 4 and 6h) on the P dissolution of the investigated HMP, HNP, MMP, MNP, LMP, and LNP samples, employing a fixed AA concentration that achieved AA/PR w/w ratio of 2:1 for each PR sample, was studied (Fig 8b). The prevailing conditions of such leaching experiments are compiled in Table 1. Careful examination of the relationship between the reaction time and P dissolution rate revealed

that raising the reaction time above 2h, has an insignificant impact on the P dissolution of both micro/nano size fractions, indicating equilibrium state attainment whatever the available $H^+$ proton in the solution. Whereas the early stage of the dissolution reaction is highly dependent on the available hydrogen ions of the acetic acid [71,75]. But with time, the contribution of $H^+$ protons became weaker. Therefore, 2h was selected as an equilibrium time. Furthermore, the outstanding P dissolution of HNP (1981–2180 ppm) followed by MNP (1375–1477 ppm), signifies the role of both particle size and gained structural disorder of these fractions [73], with some preference of the formers in accordance with its high $P_2O_5$ content (31.23 wt.%) compared to the latter (27.92 wt.%). On the contrary, the P dissolution out of the micro-fractions (HMP and MMP) fluctuate around 400 ppm over the investigated reaction time range, with HMP showing slightly higher values (ranging from 380 to 431 ppm) compared to MMP. This was correlated with particle gain sizes and the crystalline nature of these fractions and their $P_2O_5$ wt.% contents. Moreover, the coupling among the low $P_2O_5$ content (< 9 wt.%) and the high crystallinity of LMP compared to the low crystallinity of LNP explains their low P dissolution rates, which ranged from approximately 44–50 ppm and 186–198 ppm, respectively.

### 3.3.3 Reusability studies.

To assess the potential amount of dissolved P through the leaching processes of the various PRs (HMP, HNP, MMP, MNP, LMP, and LNP), five leaching cycles were executed, using a fixed AA/PA w/w ratio (2:1) as depicted in Fig 8c. The other prevailing conditions of conducting these experiments are listed in Table 1. It was revealed that the P dissolution rate of all addressed samples in both micro/nanoscales at the first leaching cycle was lowered compared to the equivalent experiments (acid concentration and reaction time). This was probably assigned to the high concentration of particles from each of these micro/nanoscales PR fractions within a fixed volume of acetic acid that diminished the solid-liquid interface due to particles agglomeration/stacking, aligning with SEM data (Fig 6). Therefore, the hydrogen protons of the acetic acid were hampered from adequate access to the surfaces of the unreacted particles [54]. Furthermore, the progressive decline in P dissolution was observed with each applied leaching cycle for all investigated samples (Fig 8c). But P dissolution continuity till the 5th leaching cycle with a remarkable performance of HNP (1495–307 ppm) and MNP (1088–253 ppm) over the other samples, validates the intervening roles of both nano scale of particles/ structural disorder upon the P solubility during the successive leaching cycles [63,76].

### 3.3.4 Other leached nutrients besides phosphorus.

Phosphate rock as a natural fertilizer, is a source of many macro- and micronutrients besides phosphorus [77]. The macronutrients P, K, Ca, Mg, and S are important for various metabolic processes as protein synthesis, chlorophyll formulation and enzyme production that are essential for plant growth [78]. They also serve as plant defense against insect attacks [78]. Whereas the micronutrients like Fe, Si, Al, Zn, and Mn, are crucial for numerous processes such as $CO_2$ fixation, synthesis of enzymes and proteins, N fixation and metabolism, as well as plant disease prevention although their absorption in trace amounts [79]. However, excessive

**Table 4. Elemental analysis.** This table displays the chemical composition of some selected nutrients in the liquidous phase after the applied acetic acid (AA) leaching process for the prepared phosphatic rocks (PRs).

| Component (ppm) | PRs | | | | | |
| --- | --- | --- | --- | --- | --- | --- |
| | HMP | HNP | MMP | MNP | LMP | LNP |
| Phosphorus (P) | 492.5 | 1991.4 | 464.5 | 1638.2 | 61.6 | 173.1 |
| Calcium (Ca) | 1643.1 | 7331.4 | 1190.5 | 4329.3 | 5769.4 | 7781.7 |
| Sulfur (S) | 168.1 | 179 | 144.3 | 263.1 | 184.9 | 364.5 |
| Magnesium (Mg) | 56.7 | 101.5 | 39.3 | 66 | 2554.9 | 2823.4 |
| Potassium (K) | 13.1 | 39.9 | 11.9 | 58.9 | 25.7 | 33.4 |
| Sodium (Na) | 56.8 | 173.2 | 50.2 | 134.2 | 30.1 | 61 |
| Aluminum (Al) | 2.1 | 2.6 | 3.4 | 4.5 | 33 | 17.2 |
| Iron (Fe) | 0.2 | 0.2 | 1.4 | 0.8 | 63.8 | 2.3 |
| Silicon (Si) | 62.4 | 848.2 | 25.7 | 370.3 | 119.5 | 677.2 |

levels of these micronutrients can be toxic. So, to evaluate the efficiency of the applied acid leaching process on the solubility of other nutrients besides phosphorus, a fixed AA/PR ratio of 2:1 (3:1.5 g, w/w), using 2h/200 rpm and 25 ml distilled water at ambient temperature, was employed for each PR sample. Like P, the concentration of these nutrients in the separated liquid phases after an adequate centrifuging process, were determined using ICP OES (Table 4). The nano-PR fractions (HNP, MNP and LNP) revealed enhanced dissolution rate of the investigated macro and micro-nutrients than micro-fractions (HMP, MMP and LMP). These results recommend the investigated samples, especially HNP and MNP in descending order, to be utilized as a high-reactive P fertilizer for direct agricultural applications. Where the slow P release can provide a constant supply to the plant along the entire growth period, leading to a better plant growth and higher biomass yield and hence more P uptake [80]. This algin with the fact that nanoparticles of PRs can be absorbed onto the clays of the soil, hampering their fixation and facilitating their release into the soil solution that can be utilized by the plant [81]. This demonstrates the commercial viability of such deposits as an alternative to chemical fertilizer. Similarly, a liquid fertilizer can be produced grounding on the leached liquid phase from both HNP and MNP fractions that contain significant amounts of phosphorus (1991.4 and 1638.2 ppm, orderly) and calcium (7331.4 and 4329.3 ppm, orderly), which are valuable nutrients for plants. However, the other nutrient concentrations may need adjustment to create a well-balanced and effective liquid fertilizer. For example, the relatively low potassium concentration from both fractions signifies that potassium may be required to make these liquid phases a well-balanced fertilizer. Also, the leached liquidous phases of LNP can be applied as Ca/Mg/S rich liquid fertilizer after conducting the appropriate balancing procedures for some of their low nutrients, with some preference of the latter fraction. Therefore, further analyses and testing are required to optimize the nutrient ratios, levels of toxic heavy metals (Cr, As, Cd, Pb, U, V and Hg) and to ensure that the produced liquid fertilizer meets the specific requirements of the target plants and soil conditions, but this is out of the scope of the current study and will be investigated in a sperate future study.

## 4. Conclusion

The outcomes of the present work can be drawn as follow:

✓ The impact of intensive grinding to nanoscale on the chemical composition, mineralogical, morphological and surface characteristics of different grades of oxidized PRs (high, medium and low-grade), were carefully investigated.

✓ Unlike LNP, the intensive grinding increased the structural disorder of HNP and MNP and raised their LOI contents (10.62 and 13 wt.%, orderly) in a surprising manner compared to their equivalent HMP and MMP (6.04 and 10.92 wt.%, respectively).

✓ The grinding process to nanoscale reduced the $S_{BET}$ and the $V_t$ of the HNP, MNP and LNP because of agglomeration phenomena that was assigned to their high LOI contents compared with their micro-scale counterparts (HMP, MMP and LMP).

✓ The P dissolution rate of the precursor micro-scale fractions and their modified derivatives all over the applied experimental parameters traced the following trend: HNP > HNP > HMP > MMP > LNP > LMP.

✓ Despite the noticeable $P_2O_5$ wt.% reduction of HNP and MNP with intensive grinding, their remarkable P dissolution rates (≈ 620–2180 ppm and ≈ 550–1640 ppm, orderly) compared to their counterpart micro-fractions, HM (≈ 173–499 ppm) and MMP (≈ 165–468 ppm), can be tied with their gained structural disorder and their nanoscale particles.

✓ The P dissolution rate of all investigated PRs at the first leaching cycle of regeneration studies was obviously reduced compared to equivalent leaching experiments of other investigated parameters (acid concentration and contact time), as a reflection of the profound role of particles agglomeration within the acetic acid solution that hampered $H^+$ from adequate access to the surfaces of the unreacted particles.

✓ The P dissolution continuity till the 5th leaching cycle with preference of HNP and MNP over the others, aligns with the fact of particle size reduction during the successive leaching cycles.

✓ HNP and MNP can be considered as a promising source of high-reactive P fertilizer for direct agricultural applications instead of chemical fertilizer. However, long-term effects of nano-sized particles on soil properties, nutrient dynamics, and potential ecological consequences should be evaluated to ensure sustainable agricultural practices.

✓ HNP and MNP can provide potential candidates for use as P/Ca rich liquid fertilizers with adequate macro and-micronutrients, whereas LNP can be counted as a source of Ca/Mg/S rich liquid fertilizer after conducting the appropriate balancing procedures for some of their low nutrients.

✓ Finally, to maximize the agronomic potential of widely distributed oxidized phosphate deposits, intensive mechanical grinding can be adapted as an eco-friendly protocol.

## Supporting information

**S1 Dataset. Raw dataset of phosphorous leaching experiments.** This data includes the effect of acetic acid concentration, effect of reaction time and reusability studies.
(XLSX)

## Author contributions

**Data curation:** Houda A. Khedr, Ahmed M. Zayed.

**Funding acquisition:** Hussah A. Alshwyeh, Najla F. Gumaah, Saedah Rwede AL-Mhyawi, Ahmed H. Ragab.

**Methodology:** Houda A. Khedr, Ahmed M. Zayed.

**Supervision:** Mohamed O. Ebraheem, Ahmed M. Zayed.

**Validation:** Houda A. Khedr, Ahmed M. Zayed.

**Visualization:** Houda A. Khedr, Ahmed M. Zayed.

**Writing – original draft:** Houda A. Khedr.

**Writing – review & editing:** Ahmed M. Zayed.

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
