## [Decision Letter · Decision Letter 0]

18 Nov 2024

PONE-D-24-36662Nanoscale Grinding: Unlocking the Nutrient Potential of Oxidized Phosphate Rocks for Sustainable Fertilizer InnovationPLOS ONE

Dear Dr. Khedr,

Thank you for submitting your manuscript to PLOS ONE. After careful consideration, we feel that it has merit but does not fully meet PLOS ONE’s publication criteria as it currently stands. Therefore, we invite you to submit a revised version of the manuscript that addresses the points raised during the review process.

We look forward to receiving your revised manuscript.

Kind regards,

Amitava Mukherjee, ME, Ph.D.

Academic Editor

PLOS ONE

2. In your Methods section, please provide additional information regarding the permits you obtained for the work. Please ensure you have included the full name of the authority that approved the field site access and, if no permits were required, a brief statement explaining why."""

Reviewers' comments:

Reviewer's Responses to Questions

**Comments to the Author**

1. Is the manuscript technically sound, and do the data support the conclusions?

Reviewer #1: Yes

2. Has the statistical analysis been performed appropriately and rigorously? 

Reviewer #1: N/A

3. Have the authors made all data underlying the findings in their manuscript fully available?

Reviewer #1: Yes

4. Is the manuscript presented in an intelligible fashion and written in standard English?

Reviewer #1: No

5. Review Comments to the Author

Reviewer #1: I read the manuscript “Nanoscale Grinding: Unlocking the Nutrient Potential of Oxidized Phosphate Rocks for Sustainable Fertilizer Innovation”, which aims to understand the effect of intensive grinding to nanoscale of phosphate rocks from Egypt.

The study involved petrographical characteristics, physical cominuation using jaw and mill, chemical, XRD, FT-IR, SEM, BET surface area analysis and leaching experiments using acetic acid. The data are excellent, and the topic is very interesting, since the results will help to increase efficiency and performance through the application of new emerging nano-fertilizers technology.

I found the English in the manuscript well-written, although I am not a native English speaker. Nonetheless, some improvements are needed to enhance the manuscript’s quality.

The main point is the effect of cominuation on the crystal size. The authors related the cominuation to the formation of amorphous and to change in the chemical composition. This topic needs be better explore with reference support. For example, I indicate the paper DOI: 10.1021/acsnano.9b05157 that shows the influence of grain size in the XRD data. The XRD does not indicate increases of amorphous phase in the samples with cominuation.

The manuscript need more support of reference in the discussion topic.

The nonconformity between chemical analyses of samples as well as FT-IR analysis in the different granulometry samples must be redone to check if it is not an analytical problem.

I added some comments in the pdf file. The green marks indicate some doubt that need improved/clarified.

A location map, with regional indication and geology is mandatory.

Put figures 2 and 3 in a single figure as well as figures 6 and 7. This will help to compare the samples characteristics. The other figures are very good.

Please do not use “&” as synonyms of “and”

Although I indicated minor revisions, a very careful revision needs to be done by the authors to increase the quality of future publication.

6. PLOS authors have the option to publish the peer review history of their article (what does this mean? ). If published, this will include your full peer review and any attached files.

**Do you want your identity to be public for this peer review?** For information about this choice, including consent withdrawal, please see our Privacy Policy .

Reviewer #1: **Yes: ** Adriana Maria Coimbra Horbe

---

## [Author Response · Author response to Decision Letter 0]

17 Feb 2025

Thanks For your time and effort; all comments were answered in the revisions files

---

## [Editor Report · Decision Letter 1]

3 Mar 2025

Nanoscale Grinding: Unlocking the Nutrient Potential of Oxidized Phosphate Rocks for Sustainable Fertilizer Innovation

PONE-D-24-36662R1

Dear Dr. Khedr,

We’re pleased to inform you that your manuscript has been judged scientifically suitable for publication and will be formally accepted for publication once it meets all outstanding technical requirements.

Kind regards,

Amitava Mukherjee, ME, Ph.D.

Academic Editor

PLOS ONE
---

## [Editor Report · Acceptance letter]

PONE-D-24-36662R1

PLOS ONE

Dear Dr. Khedr,

I'm pleased to inform you that your manuscript has been deemed suitable for publication in PLOS ONE. Congratulations! Your manuscript is now being handed over to our production team.

Kind regards,

on behalf of

Professor Dr. Amitava Mukherjee

Academic Editor

PLOS ONE